# Retroperspective:
# Learning a Structured Neural Network Policy for a Hopping Task

June 30, 2020

|                          |                                      |
|--------------------------|--------------------------------------|
| Paper written by:        | Julian Viereck, Jules Kozolinsky,    |
|                          | Alexander Herzog, Ludovic Righetti   |
| Retrospective written by:| Julian Viereck                       |

## 1 Introduction

We published the journal paper *Learning a Structured Neural Network Policy for a Hopping Task* [1] roughly two years ago as a RAL journal paper with proceedings in IROS 2018. The paper is about learning a hopping motion on a single leg robot. The paper contributes a way to learn the dynamics of the system, how to optimize a hopping policy and two different ways to transfer the optimized policy to a neural network policy. The goal of one of the neural network policies, the feedback network policy, was to learn the feedback and feedforward gains. This allows to inspect the behavior of the policy by analyzing the outputs.

In the following, I outline a few lessons learned that are not mentioned in the original paper. In addition, I am listing a few things I would do differently from today's standpoint.

## 2 Technical lessons learned

Here I outline some technical details that I became aware of during this work.

- **Experimental round trip time matters.** At some point running the simulation experiments and regenerating the plots took longer than a day. Through various optimization this time was brought down to the five minute range[1]. Faster round trip time allowed us to try new ideas and run the required experiments faster.

---

[1]Initially, most of the time was spend training very large networks with many hidden layers, neurons and large batch sizes for many steps, which we reduced to smaller networks and smaller batch sizes at no loss of final policy network performance.

- **Experiment automation pays off.** Our experiments consisted of many different tasks like first optimizing a policy, generating training data for neural network training, training different networks and then evaluating them. To orchestrate these tasks and rerun only the ones needed, we used *make* and a python generated *Makefile*. This allowed us to run all tasks including the final plot with a single command. In addition it enabled us to run tasks in parallel and thereby faster.

- **Control network policies can be small and train fast on CPUs.** During our experiments we found that the control policies we used were rather small neural networks with ten hidden neurons and three hidden layers. As the networks were small, we could train them on a CPU at similar speed compared to a GPU. This allowed us to parallelize the network training on our local workstation over multiple CPUs and speedup the overall experimental round trip time significantly.

- **Using a cluster comes with overhead.** At some points we had all our experimental simulations running on a local research cluster. While this allowed us to run more tasks in parallel, it also came at the overhead of debugging our scripts on the cluster, coordinate for free resources with other cluster users and spending time to adjust our scripts to run on the cluster. In the end, we found it easier to optimize our tasks to run on a local machine (e.g. train on CPUs instead of GPUs) and use a powerful workstation with many CPUs.

As conclusion, we found it important to have a short experiment round trip time to try new research ideas quickly. We achieved this by finding that smaller networks performed similar than the initial large ones used. In addition, we found that our small networks train fast on CPUs, allowing to utilize parallelization over multiple CPU cores when training different neural networks. Especially in our setup without vision we didn't find it beneficial to use GPUs or large compute. The parallelization was enabled by the experiment automation that we used. Last but not least we found it easier to run all our experiments locally than on a cluster.

## 3   Research lessons learned

Besides the technical lessons learned, there were also some research related aspects learned on the way.

- **Avoiding experimenter bias for real robot experiments using automation.** One important part about doing proper research is the ability to reproduce results. When it comes to real world experiments operated by an experimenter, the experimenter can bias the results of the experiments. For example, in our first first submission of the paper we changed the floor angle of attack manually. As pointed out by the reviewers, this

could lead to a bias from the experimenter and should therefore be automated. Therefore, for the final paper, we motorized the floor to have repeatable motions. This lead to a change in results and the PPO policy performed better on the real world experiment after the motorization.

- **Learning on real hardware did not work.** Initially we tried to learn the dynamics model for the trajectory optimization directly on real hardware. This did not work and we ended up using only simulated data[2]. As hypothesis, I assume the main problem for learning on the real hardware came from the cogging pattern applied by the brushless motors. Brushless motors use multiple coils and magnets and because of their alignments turning the motor has easier and harder to rotate positions. The resulting torque is relative high compared to the torque needed to move the leg in free air. Especially at the beginning of the dynamics learning this caused the learned dynamics to get biased and become unstable. Since the initial experiments we have tried to remove the cogging torque but did not manage yet. This is as the cogging torque profile is changing relatively quickly and is hard to identify.

- **Learning feedback policy using PPO.** Using the Proximal Policy Optimization (PPO) algorithm, we attempted to learn a feedback policy. That is, predict with a PPO trained network policy the feedforward torque, desired state and feedback matrix. This did not work and we hypothesized in the paper that this is due to the instability of learning the desired state and feedback matrix at the same time. Meanwhile, there has been some followup work [2, 3] which shows that a feedback like policy can be optimized when more structure is imposed on the problem (e.g. learn only a diagonal feedback matrix).

In summary, initially we missed to rule out experimenter bias for the real hardware experiments. The real world experiments got automated to remove the bias. We didn't manage to learn the dynamics directly on the real hardware due to unmodeled disturbances. Therefore, we used only simulation data to train the policies. Using PPO we were not able to train a feedback policy. This has been explored further in the meantime [2, 3].

## 4  Things done differently

Besides the technical and research lessons learned, there are also a few points I would do differently if approaching this project once more.

- **Fix network architecture earlier.** While working on the project, we spent significant time training different sized neural networks as we were

---

[2]To run the policies from simulation on the real hardware, we scaled the output torque by a factor of 1.3. In addition a small joint velocity damping $tau_{damp} = -0.03 \, \dot{q}$ was added to stabilize the motion.

not sure which one would perform best. In retroperspective, we should have picked one set of network parameters and focus more on the other research aspects.

- **Use same robot properties in simulation and real robot.** For the simulated and real robot the publication uses different total mass, maximum torque etc. This is the case as the real hardware was finished later. We decided to not adapt the robot parameters in simulation to show the learned policies transfer to the real robot and thereby show the robustness property of the policies. From today's standpoint, I would rather use the same robot properties in simulation and on the real hardware. This is to make the results in simulation and real robot better comparable. In addition, the transfer to a different robot platform was not the main focus of the publication.

- **Use metric directly corresponding to the task.** As metric for our hopping task we used the sum of the upwards base velocity squared. This metric showed well the progress when optimizing trajectories with iLQR over multiple jumps and while the trajectories got longer over the course of optimization. However, naturally a jump is better the higher the robot manages to jump. Therefore, I would use a metric like the maximum mean jump height over multiple jumps instead today.

- **Use appropriate contact model.** In our work we used a momentum-preserving contact model without slippage. We choose this contact model to simulate hard impacts like we observe them on the platform properly. Because the model didn't support slipping, the simulated robot leg would not slip when the floor was at a very steep angle. However, simulating slipping in this scenario is crucial to have a realistic simulation. Therefore I should have used a contact model allowing foot slippage, like a spring-damped contact model with friction cone for the tangential forces.

In conclusion, we spend significant time training different neural network architectures and should have converged on an architecture earlier. We used different robot properties for historical reason. We should have updated the simulated properties to make the experiments more comparable. As overall metric, we used a metric that helped us to see the progress of the iLQR optimization but should have used a metric more corresponding to the task instead. Last but not least, as slippage is important for our task we should have chosen a contact model that allows to simulate slippage as well.

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
