# OpenReview forum: "Learning a Structured Neural Network Policy for a Hopping Task"
_roboticsfoundation.org/RSS/2020/Workshop/RobRetro — RobRetro 2020_

### Official Review · AnonReviewer1 · 2020-06-19
**Good retrospective about technical and research details of learning on robotics hardware. Some more details would be good in the final version.**

**Rating:** 8
**Confidence:** 5

**Review:**

This paper presents a retrospective on a published work on learning hopping policies. The paper provides interesting realizations, during and after the submission, as well as insights into the research and design choices made. I think this is a good fit for the workshop, but some additional details will clarify the points made in the paper better. My points of confusion are below:

Section 2:
1. What were optimizations that brought down experiment time from >1day to 5 minutes?
2. Are robotics experiments slower on GPUs partially because most simulations are on the CPU, leading to an additional overhead of data transfer from CPU to GPU?
There is an interesting insight here that robotics experiments that do not use vision do not benefit from GPUs and large compute. It would be good if authors could make a concluding remark for this section, that summarizes these findings.

Section 3:
1. Interesting insight about dynamics learning on hardware being hard, either due to noise, or unmodeled disturbances. I am surprised that dynamics learning in sim transferred to hardware. Were there other tricks in getting this to work?

Section 4:
In general, this Section mentions choices made in the past, and how you would do this differently. Could you motivate the reason for the past choices? For example:
1. Why was the sim dynamics properties chosen to be different from hardware?
2. What was the reason for using velocity reward? Are there advantages/disadvantages to the two choices? I can imagine that velocity is easier to measure on hardware, and final height harder? Is one of the two sparse reward, and the other dense?

In general, very interesting retrospective. Some more details would be great.

---

### Decision · Program_Chairs · 2020-06-25

Accept